# Negative Perception of Bats, Exacerbated by the SARS-CoV-2 Pandemic, May Hinder Bat Conservation in Northern Uganda

Imran Ejotre [1,2,*], DeeAnn M Reeder [3], Kai Matuschewski [1], Robert Kityo [4] and Juliane Schaer [1,*]

1    Institute of Biology, Humboldt Universität zu Berlin, 10115 Berlin, Germany
2    Department of Biology, Muni University, Arua P.O. Box 725, Uganda
3    Department of Biology, Bucknell University, Lewisburg, PA 17837, USA
4    Department of Zoology, Makerere University, Kampala P.O. Box 7062, Uganda
*    Correspondence: ejotreim@hu-berlin.de (I.E.); schaerju@hu-berlin.de (J.S.); Tel.: +49-30-2093-6062 (J.S.)

**Abstract:** Bats face diverse challenges that cause global bat population declines, including habitat loss and roost disturbance. Additionally, negative perceptions of bats and their potential role in several zoonotic diseases have led to actions against bats. We documented existing knowledge and perception of bats through interviews with 151 participants of fifteen tribes in Northern Uganda in 2020 and conducted a sensitization campaign that prevented planned actions against bats. The interviews revealed distinct firm beliefs, negative perceptions, limited knowledge on bats, and the influence of media in shaping actions against bats. In addition, modified landscapes and habitat loss increased encounters and subsequent deterioration of relations between humans and bats. Targeted threats towards bats were exacerbated by public misinformation during the SARS-CoV-2 pandemic. No deliberate conservation efforts exist, and negative perception largely hampers the implementation of bat conservation in Northern Uganda. Importantly, the study also demonstrates that sensitization campaigns can be effective tools to protect bats in the short term. Regular sensitizations and education are recommended for sustainable changes in attitudes to and coexistence with bats.

**Keywords:** Uganda; bats; indigenous knowledge; perception; conservation; education; human-bat interaction; public misinformation





## 1. Introduction

Bats have a poor reputation in much of the world, most recently due to the putative origin of several zoonotic viruses in this diverse group of flying mammals [1–8]. Fear of contracting dangerous diseases has even led to killings of bats in several countries [9]. Although some particular bat species have been confirmed as sources of some zoonotic diseases, including Marburg virus in Africa and Nipah virus in South-East Asia [6,10], misleading and unclear scientific and political communication during the SARS-COV-2 pandemic has added to the misery of bats [11]. In addition, bats face myriad threats that have caused some significant bat population declines worldwide [12,13], including loss of habitat [14–16], roost disturbance due to hunting and guano mining [17–19], and invasive pathogens (e.g., the fungus that causes white-nose syndrome (WNS)) [20]. Indeed, of the ~1456 species of bats [21], 24% (18% Data Deficient (DD)) are currently considered globally threatened [22]. These challenges demonstrate the urgent need for enhanced and focused conservation efforts for bats, the success of which also depends on the attitudes of people towards bats in a given area.

In many parts of the world, local myths and traditions portray wildlife as potentially dangerous animals [9,23–28]. Negative perceptions largely overshadow any understanding of the beneficial ecological roles of bats [23,26] and result in negative attitudes, reduced empathies, and direct persecution of bats in many regions [24,29].

In Northern Uganda, no initiatives for local bat conservation exist, and human knowledge, attitudes, and perceptions about bats in this biodiverse country have not been

documented. Nearly 97 of 330 recorded mammal species in Uganda are bats [21,30] and yet bat diversity surveys have been conducted in very few areas and the distribution of bats in the country remains largely understudied [31]. Accordingly, ecological information on bat diversity, assemblages and abundance in Uganda and predictions on bat-human interactions are urgently needed. In addition to animal diversity, Uganda is also culturally rich, with 65 indigenous communities that vary in language, values and cultural norms [32–37]. Indigenous knowledge is under threat and changing in response to modern pressures [36–39]. Our objective in this study was to document existing knowledge and perceptions of bats in Northern Uganda among different local communities and to describe differences in viewpoints and knowledge between elders and younger study participants. Additionally, we present the results of a sensitization campaign about bats that efficiently mitigated COVID-19-related persecution of bats.

## 2. Materials and Methods

### 2.1. Study Area and Demographic Information of Participants

The study was conducted during the months of June, July, August, and October 2020 in the West Nile region of north-western Uganda, which borders South Sudan to the north and the Democratic Republic of Congo to the west (Figure 1A). The Nile River separates the West Nile region from greater northern Uganda to the east and the confluence of Lake Albert to the Nile River in the South. The area is characterized by savanna mosaic along the Nile River at elevations of about 600 m above sea level (asl) but rises steadily towards the west to elevations of over 1300 m above sea level with features of riparian forests [31]. Fruit trees, such as mangoes (*Mangifera indica*), are a common feature throughout the study area, making the presence of fruit bats very common. Despite their huge representation in Uganda's mammalian fauna, bat diversity surveys have been conducted in very few areas and the distribution of bats in the country is poorly known [31,40] and no data has been published from the West Nile region to date.

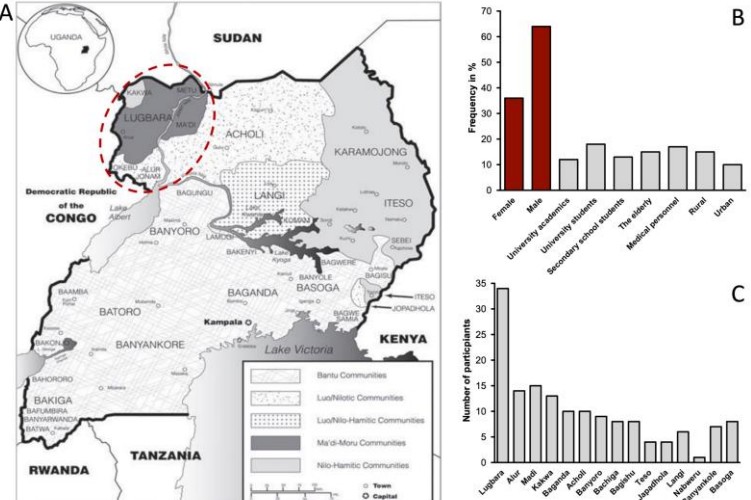

**Figure 1.** (**A**) Distribution of Ugandan communities (adapted from source: Minority Rights Group, 2001). The study area in the northwestern region is highlighted with a red circle, (**B**) Demographic information (number of participants: *n* = 151) on participants in every category is given in percent. (**C**) Number of participants per Ugandan tribal group. Note that not all tribes are geographically located in the study area. Note, "Buganda" = tribe name, but a "Muganda" is a person from the Buganda tribe (singular), while "Baganda" are people from the Buganda tribe (plural). The same terminology applies for all Bantu tribes whose names begin with 'B' (e.g., Bunyoro and Munyoro and Banyoro; Bugisu and Mugisu and Bagisu; Busoga and Musoga and Basoga, etc.).

Study participants (local people who live in the communities) were recruited from public institutions, i.e., Muni University, Secondary schools, Arua Regional Referral Hospital, private health facilities in Arua City and Koboko, Yumbe and Moyo districts, and from random encounters with people living in urban and rural areas in the wider study area (Figure 1A). Overall, 151 people participated in this study, 60 in individual interviews and 91 in a group setting. Gender, age, occupation, and tribal community were recorded for every interviewee. A representation of the overall demographic of the region, gender, age cohorts (defined as young (<20 years), adults (20–49 years) and elders (≥50 years)), and communities in urban and rural areas was aimed for (Table 1). Elderly people were defined as ≥50 years based on the Ugandan population pyramid 2022 [41]. In rural areas, it was challenging to carry out individual interviews or gather specific groups, in these cases, groups comprised members of different ages. Interviews were conducted among university academic staff and students, secondary students, medical personnel, the elderly in the community, persons from rural areas and urban settings (Table 1). Male respondents (64%, 96/151) outnumbered female respondents (36%, 55/151) despite our efforts to obtain equal representation of gender in the study, mainly due to the heavy daily routine of the women (Figure 1B, Table 1). Study participants comprised members of 15 indigenous communities across Uganda. The main communities within the study area include the Lugbara, Alur, Madi, and Kakwa speaking people although each of these tribal communities has dialectical groups within it [42,43] (Figure 1C). The groups of the Baganda, Acholi, Banyoro, Bachiga, Bagishu, Teso, Japadhola, Langi, Nabweru, Banyankole, and Basoga were represented through members living in the region or working in institutions located within the study area (Figure 1A,C). As the survey was conducted during the SARS-CoV-2 pandemic, safety measures were implemented to minimize pathogen transmission, such as keeping physical distance of at least 1.5 m, sanitizing frequently, and wearing medical masks. Hand sanitizer and masks were provided to participants. Interviews were carried out as individual or group interviews for specific cohorts (e.g., university academic staff, university students, secondary school students, the elderly, medical personnel, rural community).

**Table 1.** Overview of participants for interviews of local knowledge and perception of bats.

| | Category (Number of Participants) | Age in Years [2]/Mean age (Range) in Years [3] (Number of Participants) | Gender (Women/Men) |
|---|---|---|---|
| **Individual interviews** (*n* = 60) | University academic staff (*n* = 10) | <20 (0), 20–49 (10), ≥50 (0) | 0/0, 3/7, 0/0 |
| | University students (*n* = 12) | <20 (5), 20–49 (7), ≥50 (0) | 2/3, 1/6, 0/0 |
| | Secondary school students (*n* = 4) | <20 (4), 20–49 (0), ≥50 (0) | 2/2, 0/0, 0/0 |
| | The elderly (*n* = 7) | <20 (0), 20–49 (0), ≥50 (7) | 0/0, 0/0, 1/6 |
| | Medical personnel (*n* = 6) | <20 (0), 20–49 (5), ≥50 (1) | 0/0, 1/4, 1/0 |
| | Rural community (*n* = 6) | <20 (0), 20–49 (3), ≥50 (3) | 0/0, 0/3, 0/3 |
| | [1] Urban community (*n* = 15) | <20 (0), 20–49 (15), ≥50 (0) | 0/0, 6/9, 0/0 |
| **Group interviews** (*n* = 12) (total of 91 participants) | University academic staff (1 group, *n* = 8) | 38 (33–55) | 2/6 |
| | University students (2 groups, *n* = 15) | 21 (19–27) | 6/9 |
| | Secondary school students (2 groups, *n* = 16) | 17 (16–19) | 8/8 |
| | The elderly (2 groups, *n* = 15) | 58 (50–78) | 6/9 |
| | Medical personnel (3 groups, *n* = 20) | 36 (26–60) | 12/8 |
| | Rural community (2 groups, *n* = 17) | 32 (18–72) | 4/13 |

[1] Participants outside of institutional settings in urban environment. [2] Age in years for individual interviews. [3] Mean age (range) in years in group interviews.

## 2.2. Individual Interviews

A semi-structured questionnaire about the "Knowledge and awareness about bats in Northern Uganda" was developed. The English language questionnaire comprised eight main questions and six additional questions for selected interviewees (Supplementary Materials Text S1). Interviews lasted approximately 15 min. A guide read out the questions to the interviewees. The questionnaire was not translated into the different tribal languages; instead, translators were engaged when needed. In most interviews, the first author (IE) was the translator for the dominant languages spoken in the study area. Prior to the survey, the questionnaire was pretested by interviewing 35 persons and two focus groups each to harmonize and refine the questionnaire by the research team. To keep the interview time short, the additional six questions were targeted to selected respondents on their experience in life (the elderly). The interviewer recorded all information on the printed questionnaires. At the end of every interview session, time was given for the interviewees to ask any questions or raise any queries about bats. The extra time also provided opportunities to sensitize the respondents to the vital ecological roles that bats play in nature and how best to coexist with them within our communities.

In documenting prevailing stories about bats in each community, we asked one open-ended question "What stories does your community have about bats?". This allowed the respondents to freely speak out about anything they have heard about bats in their community (Text S1). To assess the willingness of the communities to coexist with bats, we asked the following questions: "Should bats be conserved?"; "If yes, what suggestions do you propose that can best work to conserve bats in your community"; and "If no, what are the reasons for your position?". When the issue of bats as a cause of diseases was raised as one reason to not conserve bats, follow-up questions were asked: "Examples of diseases allegedly spread by bats.", "Where do people get information about bats and diseases?". To understand how people related with bats in the past and how this relationship has evolved over time, a complementary investigation targeting the elderly was conducted.

## 2.3. Focus Group Interviews

Focus group interviews were held on the basis that: (1) participants were more likely to consent to participate in a group setting; (2) the group discussions setting facilitated contributions and thus information; (3) movement of both the interviewers and potential participants was limited during the pandemic lockdown and group interviews allowed involvement of more participants in each area.

A total of 12 group interviews with 91 participants were carried out (Table 1), numbers of women/men were recorded, and the average age of the group was estimated by sensitively approaching and asking the youngest, oldest and one member in between about their age. Of note, most people did not want to share their age information (Table 1). The interviews were conducted as informally as possible to allow participants to freely express themselves and not withhold information. The interviewer subsequently provided more information and sensitized the participants only after the participants had exhausted their views on the matter. Focus group interview sessions typically lasted up to sixty minutes.

## 2.4. Rapid Conservation Action

The documentation of indigenous knowledge and perception of bats in Uganda was accompanied by a rapid community sensitization program that was created ad hoc after the leading author (IE) became aware of several planned actions against bats and bat roosts in the study area. The community sensitization program, conducted in October 2020, focused on the importance of bats and how people can safely coexist with bats. Information was offered to counter the narrative that attributes every dangerous emerging disease to bats. We used printed copies of the booklet "Living Safely with Bats" that was developed by EcoHealth Alliance and partners as a resource to educate participants on ways to mitigate zoonotic disease spillover risk [44,45]. We received permission to adapt the booklet (adding pages) to address the specific issues of the communities in Northern Uganda. We further

compiled information about bats, e.g., diversity, ecology, diseases, from publications that were presented to the communities. The sensitization study comprised a (1) presentation followed by questions and answers, (2) the distribution of the booklet and (3) in-depth conversations with the communities, community leaders, and members of institutions whose premises have colonies of bats.

### 2.5. Data Analysis

Data from individual questionnaires and focus group discussions were quantified. Due to the low and heterogenous sample sizes, data are presented using descriptive statistics, with total numbers or proportions/percentages reported. Some folktales or community-specific stories were written as narrated (verbatim) to maintain the originality. Most were narrated in local languages and translated to the best understanding of the translator.

## 3. Results

### 3.1. Basic Knowledge of Bats

Almost all respondents in the individual interviews reported the existence of bats in their area (58/60) and described up to four different "types" of bats (Figure 2, Table S1(A,B)). The descriptions of bats were mostly based on color (brown, black, grey, etc.) or size (big, small, medium) (Figure 2, Table S1(B)). Few respondents described the shape or other features of bats, while others described bats based on diet (e.g., fruit-eating or insect-eating bats), or where they roost (e.g., house bats, cave bats, banana bats, latrine bats, tree bats, etc.). These descriptions were similar in the group interviews (Table S1(B)).

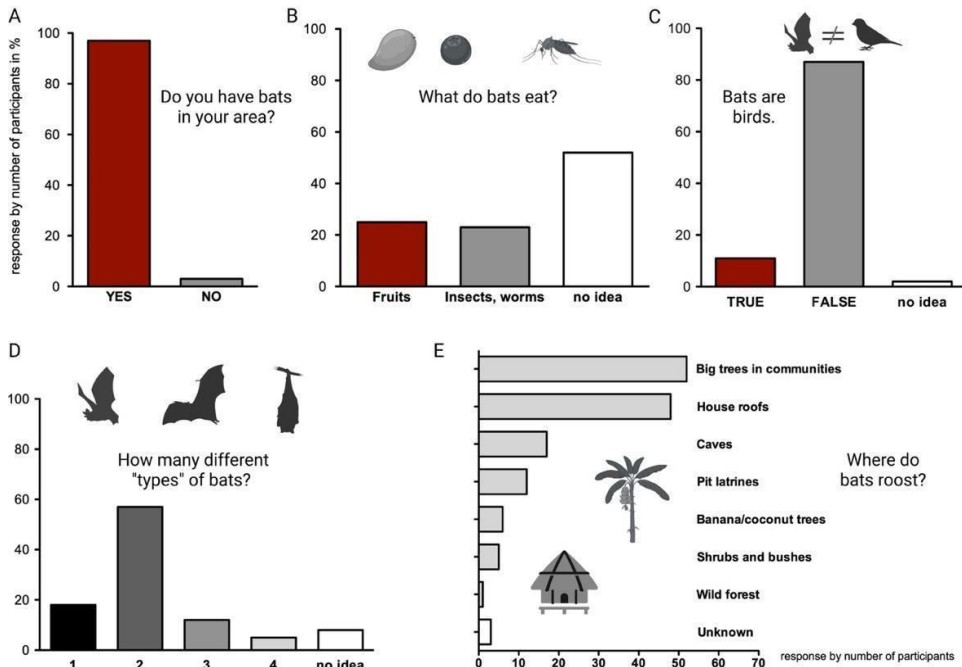

**Figure 2.** Overview of knowledge about bats from individual interviews (*n* = 60). (**A–D**) Response by number of participants is given in percent. (**E**) Response by number of participants is given as number, since multiple answers were possible. (**A**) Presence of bats in the study area was acknowledged by almost all participants. (**B**) Limited knowledge about the food sources of bats. (**C**) The large majority of participants knew that bats are not birds. (**D**) Most respondents could identify up to two different types of bats. (**E**) Respondents named up to seven different spaces bats utilize as roost sites. Most frequently mentioned roosts included big trees in communities, house roofs and caves.

Most respondents knew that bats were not birds (Figure 2, Table S1(A)). The respondents who could not differentiate bats from birds, included both males and females mostly in urban settings, e.g., housewives, traders, and students. From those who could dif-

ferentiate bats from birds, the most known difference between bats and birds was that bats give birth to live pups while birds lay eggs, followed by bats having teeth, while birds have a beak. Other differences mentioned included fur in bats vs. feathers in birds, that bats are nocturnal vs. birds are diurnal, and that bats hang upside down vs. birds sit upright. When asked about the places in their area where bats roost or bats can be found, respondents in both the individual and the group interviews named house roofs and big trees in homesteads as the most likely spots, while some respondents knew about caves, pit latrines, banana/coconut leaves, wild forests, and crevices as bat roost locations (Figure 2, Table S1(B)).

### 3.2. Benefits of Bats and Willingness of Local Communities to Protect and Conserve Bats

Next, the respondents were asked whether bats benefit humans. A small majority of the individual interviewees agreed (Figure 3) and only few participants in the group interviews said that bats benefit humans. Some university and secondary school students and rural community members stated that bats do not benefit humans (Supplemental Table S1(A)).

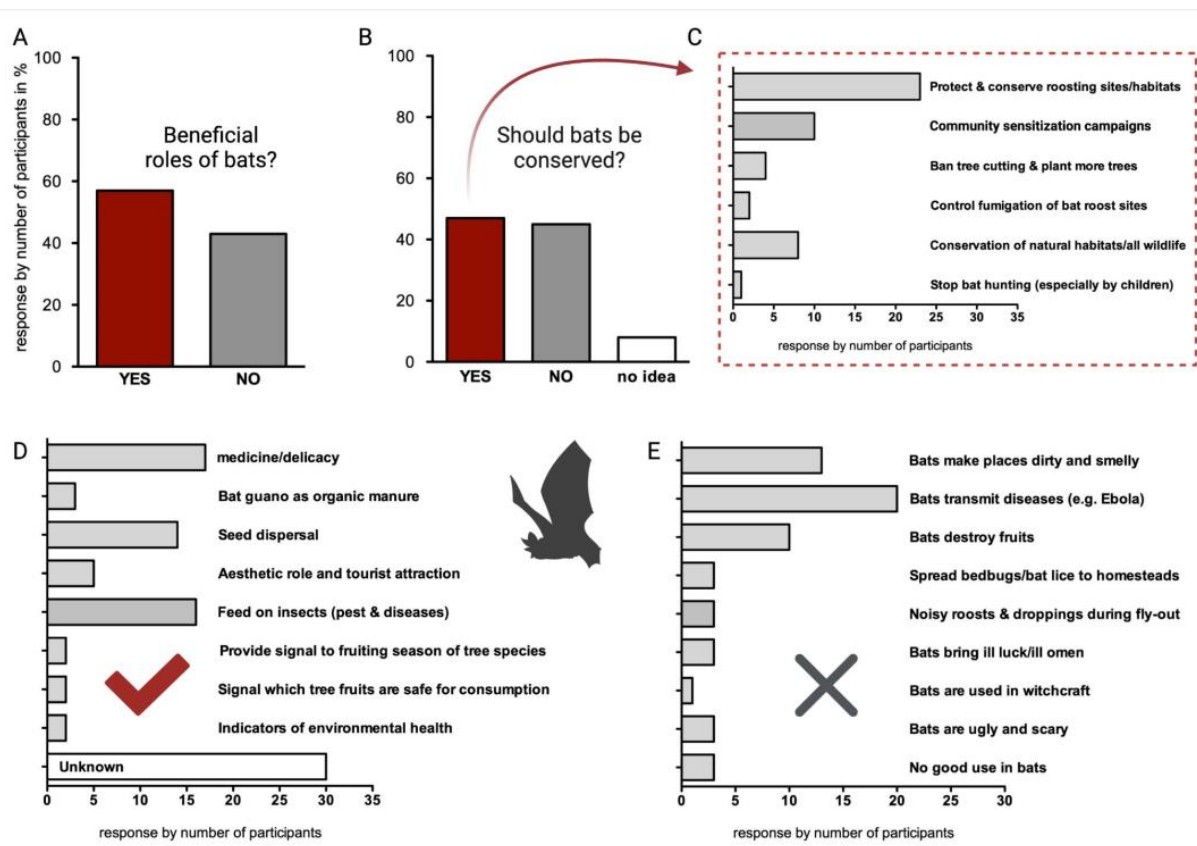

**Figure 3.** Overview about knowledge of benefits of bats and willingness of local communities to protect and conserve bats from individual interviews (*n* = 60). (**A,B**) Response by number of participants is given in percent. (**C–E**) Response by number of participants is given as total number. (**A**) Number of respondents that acknowledged that bats play beneficial roles in the environment was slightly higher than those who disagreed. (**B**) Equal proportions of participants stated that bats should be conserved or not. (**C**) Bat conservation measures proposed by participants that found that bats should be conserved. (**D**) Beneficial roles of bats listed by respondents. (**E**) Negative roles of bats listed by respondents.

Respondents were further asked about the uses and benefits that bats provide in their community, and the most common answers were that bats feed on insects that carry diseases and destroy crops, bats are important for seed dispersal and they have medicinal use (Figure 3D, Table S1(B)).

About half of the interviewees agreed that bats should be conserved (Figure 3, Table S1(A)). In a subsequent question, participants were asked, if bats should be protected, what actions should be taken in their community. Respondents mainly replied that the bat roosting sites and habitats must be conserved, sensitization campaigns are necessary and natural habitats should in general be protected to conserve all wildlife (Figure 3, Table S1(B)). A common belief was that natural environments/bat habitats should be protected, with the hope that bats will use the forests/big trees instead of human habitation (Table S1(B)). The respondents that argued that bats should not be conserved were asked for the underlying reasons and the majority argued that bats transmit many diseases like Ebola and COVID-19. Others felt that bats e.g., make places dirty and smelly and destroy fruits (Figure 3E, Table S1(B)). According to the descriptions by the respondents, people are bothered by fruit bats in trees (most probably epauletted fruit bat species, e.g., *Epomophorus* spp. and *Eidolon helvum*) and insectivorous bats that roost in their houses (most probably molossid bats).

### 3.3. Community Relations with Bats over Time: The Perspective of Elders

We targeted the elderly to understand how people in the study area have related to bats in the past. Detailed interviews were held with 16 participants from the four main tribes located in the study area: Alur, Kakwa, Lugbara and Madi. The overall narrative was that, in the past, there were plenty of natural landscapes, such as dense forests, that provided habitat for wild animals. Human settlements in Northern Uganda were scattered homesteads with few people. Wildlife habitat and the homesteads did not overlap and contact between bats/wildlife and humans was rare; thus no particular attention was paid to the bat populations. According to the interviewees, the relationship between bats and humans has changed over time. The majority of the interviewees (12/16) observed a dramatic increase in the presence of bats in human communities over the last two decades. Agriculture, deforestation, population growth and encroachment were named as triggers for this change. Over time, bats posed a problem as they are blamed for the destruction of agricultural fruits, and are noisy and smelly. In addition, all participants had some information about the association of bats with diseases, such as Ebola, Marburg, Rabies, and COVID-19. Sources of information about bats as disease spreaders included: social media platforms, FM radios, internet, family and relatives, community leaders, and members in the community (Figure 4, Table S2). To the question of how people intend to relate with bats in the future, the popular position of the elderly respondents was that bats and humans should not be allowed to stay in proximity and bats should be eradicated from human settlements (Figure 4). In general we found that older persons shared more detailed stories and local knowledge about bats.

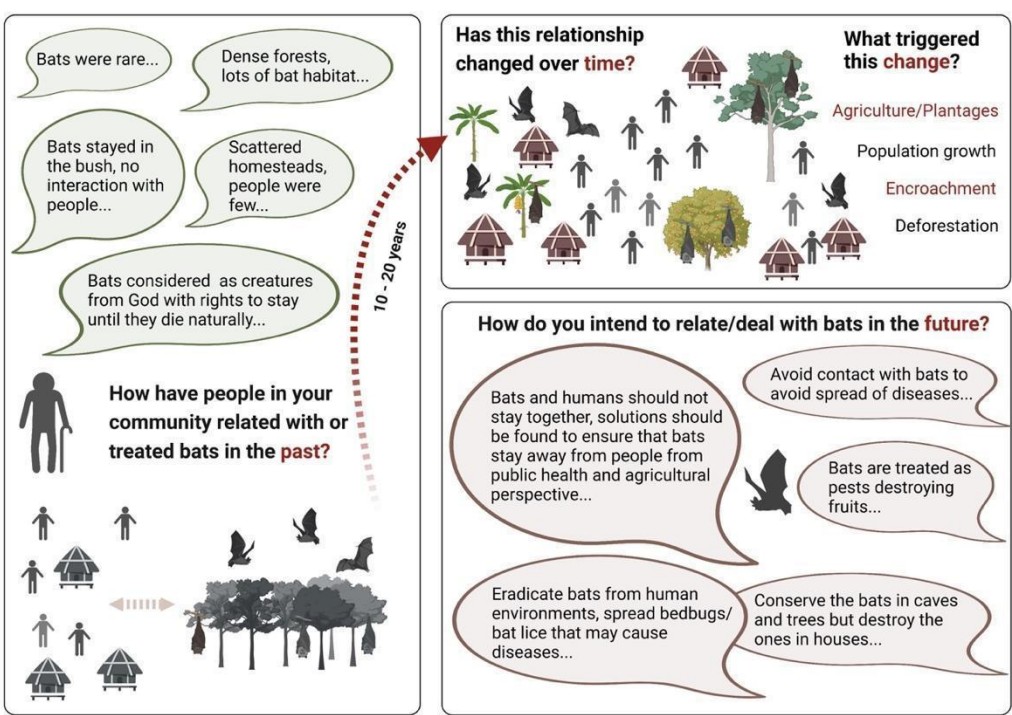

**Figure 4.** Overview of information gathered from the elderly participants (age ≥ 50 years) from the extended interviews (*n* = 16). Left box: In depth interaction with representatives of the increasingly scarce age group of >50 years on how communities related with bats in the past generated various responses summarized in the box. Box top right: Change in human relations with bats observed in the past 10–20 years, where the presence of a higher number of bats in human-dominated spaces has been reported. The main triggers of this change were described as agriculture, population growth/settlements/urbanization, encroachment, and deforestation. Box bottom right: Responses on how the participants intend to relate with bats in the future. Most solutions are based on separation of spaces between humans and bats.

### 3.4. Myths, Local Stories, and Bat Perception among the Ugandan Tribes

Both the individual and the group interviews revealed that, within the same geographical location, members of different tribes shared slightly different stories and narratives about bats. Folktales were registered from respondents that belong to the tribes of Buganda, Acholi, Bachiga, and Madi. The respondents of the other tribes did not report folk stories although respondents of every tribal community that participated had a set of myths or beliefs about bats. Interviewees of the Lugbara tribe related that bats are messengers of death and used in witchcraft, and reported an ignorance of bats across all age groups. Respondents of both the Alur and Acholi (Luo ethnic group) related that bats are wise animals, acting both as mammals and birds. When the environmental conditions are harsh for mammals, bats are birds and vice versa. Other respondents (Acholi Lamogi community) further felt that eating bats makes people very intelligent and some respondents (Alur and Acholi) shared the myth about bats transforming into vampires that suck human blood, a myth that still leads to fear in children. A middle-aged (45 years old) man, a Muganda and a university lecturer told their local stories about bats:

> "It has been related that, in our culture, if bats enter your house before construction is complete, you will never complete that house. To avert this misfortune, one should close the upper parts of the house before bats enter the house. Bats are also not in the good books of our traditions because they are believed to have defaulted their tax obligations to the king of Buganda. Folktale has it that all animals used to pay tax to the Buganda king, but the bat played tricks to evade tax payment. Each time taxes were being collected from land animals, the bat claimed to be a flying animal (bird) and when the turn to collect

*taxes from birds and other flying animals came, the bat considered itself a mammal with teeth and so different from birds. Because of this cunning trait, bats were regarded to have no respect for the king, don't contribute to development and therefore no need to conserve them".*

The man continued to state that no Muganda with strong traditional attachments has a positive attitude towards conservation of bats. A few respondents (of the Bayankole and Bachinga) believe that bats are used by witches to practice witchcraft and that the appearance of a bat foretells death of a person in a community. The respondents further fear the bats as they think that bats kill cows and humans by sucking blood from the ears at night and believe that bats bring bad omen to a home and community. The following were the cross-cutting beliefs and myths across respondents of almost all tribes represented in the study: bats are signs of bad/ill omen, bats are used in witchcraft, bats cause diseases (like Rabies, Ebola), bats are vampires that suck blood of animals and humans at night, bats are messengers of death. An important observation was the fact that not only bats, but also other nocturnal animals such as owls, are negatively associated with bad omen/luck, messengers of death, use in witchcraft, etc. Overall, bats are associated with negative attributes among all investigated tribes. This is also reflected in the totems of the many clans. Each clan in Buganda for example has a totem that mostly represents an animal; however, no totem of any Ugandan clan/tribe features a bat.

*3.5. Sensitization Campaign and Intervention of Actions against Bats*

Specific actions against bats were planned in the communities in Koboko, Yumbe and Moyo districts of the study area around October 2020 as people were concerned about the links between bats and the SARS-CoV-2 pandemic. People started to actively chase away bats from their homesteads, surrounding areas and marketplaces. Mainly colonies of fruit bats (*Epomophorus* spp., *Eidolon helvum*) were targeted with this approach to get rid of bats from shade trees close to homes or frequently used by humans. In Moyo district, the COVID-19 task force led by the District Health Officer had planned to fumigate large roosting colonies of the fruit bat *E. helvum* on a main road, where communities had multiple negative experiences with the bats (Text S4). Similarly, the management of the hospital in Yumbe district had decided to either fumigate all bats (*Epomophorus* spp.) in the mango trees or cut down all trees on the hospital premises. The shades of the mango trees are used by outpatients and inpatient attendants as resting places. When the origin of the SARS-CoV-2 virus was associated with bats, people were scared about sitting directly below bat roosts and feared getting the virus via aerosol transmission. During the campaign, the lead author (IE) used the booklet "Living Safely with Bats", which had been adapted to the local environment and presented information about bat diversity, ecology, diseases and how to live safely with bats to members of the communities (Figure 5A). The impression was that people were curious, and the presentations were followed by several questions, e.g.,: "How safe are we now as we have been staying close to these bats for some time?", "If bats are not confirmed to be the direct source of coronavirus as you have stated, why should we believe you and not the information we receive from radios and people?". Several in-depth conversations took place with the communities, community leaders, and members of institutions whose premises have colonies of bats (Figure 5). Generally, the sensitization campaign was well received and attended (see Text S4 for campaign details). No incidences of hostility towards the sensitization team were registered, and the curiosity of the participants through questions were handled with the evidence available including giving them copies of the booklet. As a result of the sensitization campaign, the planned fumigation in Moyo was temporarily halted and the cutting of trees at the hospital in Yumbe was not carried out. In Yumbe, tents were set up under the mango trees for outpatients and visitors to prevent direct contact of people with bat urine/feces (Figure 5B). In Moyo, the council did not approve the supplementary budget for fumigation and resolved to not allow destruction of bat roosts until more convincing new evidence that bats are responsible for the spread of the pandemic virus.

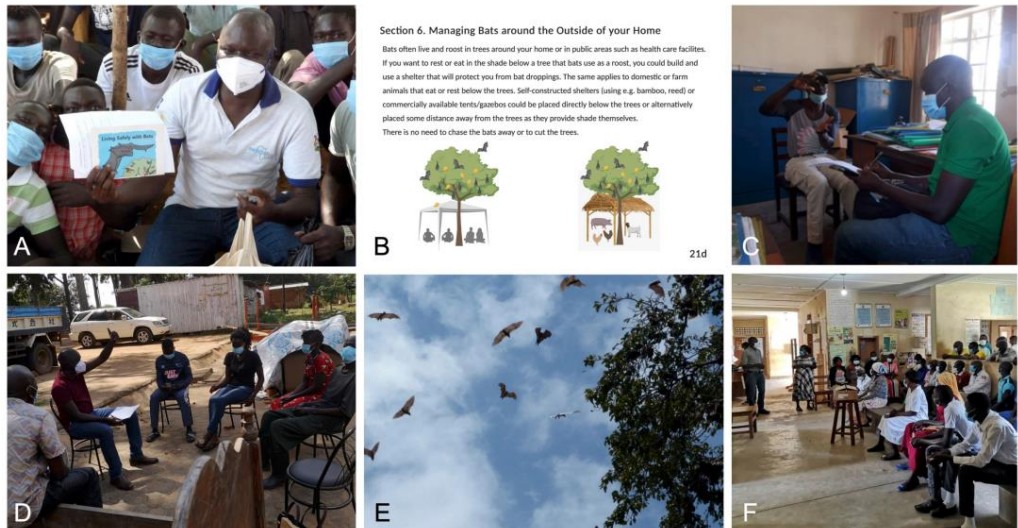

**Figure 5.** Sensitization campaign in Northern Uganda. (**A**) Depicted is the leading author (IE) showing the cover page of the booklet "Living Safely with Bats" which was reprinted and adapted from [44,45] with permission from EcoHealth Alliance and the PREDICT consortium, (**B**) Example of additional booklet page added by the authors of the study to address the specific issue of the communities in Northern Uganda, how to safely use space under bat roosting trees, (**C**) An interview on bats with a district official, (**D**) Interactions with community members, (**E**) Emergence of fruit bats (*Eidolon helvum*) from a colony at dusk in Moyo, (**F**) Sensitization and dialogue with the medical staff and management of Midigo Health Facility to spare the bats that roost in the medical facility and trees on the facility compound. Verbal consent to photograph participants was granted.

### 3.6. Limitations and Potential Bias of the Study

This study was designed ad-hoc and carried out promptly after negative actions against bats in the study area were observed during the SARS-CoV-2 pandemic. We focused on the experiences and stories of people with and on bats to gather information that can serve as a baseline for future studies. Carrying out the study during the pandemic with travel restrictions and new rules of conducting business posed several challenges for gathering information through interviews, group discussions, and sensitization efforts.

## 4. Discussion

### 4.1. Participants' Knowledge on Bats

We found that knowledge on bats was limited across communities and ages. Similar to results in other studies (e.g., [26,46]), the majority of the respondents was aware of the existence of bats in their area. However, most participants only knew one or two different types of bats which is in line with a study in Madagascar [26], while a few named four types as also reported in a study in Ghana [47]. Most participants had not seen a bat at close range to give a concise description of the animals, but the types of bats were mostly defined by body size and color which is consistent with other studies (e.g., [26]). The study did not specifically measure the influence of the level of education on the knowledge of bats, however, approximately half of the respondents were students or employees from institutions, such as universities and health facilities and no noticeable difference in knowledge was apparent. The content of animal biology taught in school curricula primarily focuses on animal identification and little to no emphasis is placed on their ecological roles and benefits to human society. Youths, especially those raised in urban settings, rarely seek opportunities to explore and experience the natural environments, thereby limiting their knowledge on wild environments to their education. Moreover, a large proportion of young people in the study area only have primary level education, as few secondary schools are supported by the government. Apparent gaps in the school

curriculum can be targeted by sensitization campaigns, which include learning material for classrooms. Because families have handed over the processes of learning, education, and transfer of knowledge to schools, the transfer of unique forms of indigenous knowledge that were previously transmitted informally is jeopardized.

Rural people derive much of their sustenance directly from natural environments, and almost all the elderly respondents grew up and/or experienced living in rural villages. This may explain why some elderly respondents and people living in rural villages were cognizant of the fact that bats play a particular beneficial role in their area: "bats provide signal to the fruiting and ripening seasons of different tree species, and that bats also give signal to humans on which fruits are safe for human consumption explaining that fruits eaten by bats have always proven safe for humans." Perhaps plenty of such unique knowledge still exists with the few remaining elderly people scattered in the rural areas that our study could not reach. Uganda's population is very young. Up to 75% of Ugandans are below the age of 35 [41]. The population above 70 years old, which probably holds more critical indigenous knowledge about bats, constitutes less than 5% and is scattered in the countryside. Before such a knowledge base is completely lost, there is a need to document it, and further studies are warranted to systematically source local expertise, including for conservation efforts.

### 4.2. Participants' Perception of Bats

Only a small majority of respondents held positive attitudes towards bats, while other studies in Africa and Asia report larger proportions of respondents with positive values [46,48,49]. The following six negative attributes assigned to bats were the cross-cutting beliefs and myths across the respondents of almost all the ethnic groups represented in the study: (i) bats are ugly and scary, (ii) bats are signs of bad/ill omen, (iii) bats are used in witchcraft, (iv) bats cause diseases (like Rabies, Ebola), (v) bats are vampires that suck blood of animals and humans at night, and (vi) bats are messengers of death. Interviewees also associated these negative attributes to other nocturnal animals. It is tempting to speculate that the nocturnal lifestyle of bats severely limits the understanding most communities have about these animals. The knowledge gaps are filled with myths, tales, and illusions that together contribute to achieving certain societal norms especially those intended to scare children to keep away from some animals or to conform to certain rules of society [23,27,28]. It also aligns with the disease avoidance hypothesis which proposes that knowledge, beliefs, and myths that associate neutral objects with disease contamination make such objects disgusting and people try to avoid such objects [29,50]. The association of bats with danger might be related to the relative lack of information regarding potential pathogens that are or might be transmitted by particular bat species. Because lack of information about bats spans across ordinary people and scientists, it is important to first improve communication amongst scientists working on bats to circulate accurate information and aim at developing consensus messaging within the scientific community to avoid passing contradicting information to the public [51]. Providing accurate information on bat biology and pointing out that the viral zoonotic risk among bat hosts is not higher than among other mammalian and avian reservoir hosts [52] might help the public to better understand the fascinating biology of bats and their vital roles in ecosystems on the one hand, and their disease transmission potential on the other hand.

### 4.3. Community Relations with Bats over Time

The most common response from the survey question "how have people related with bats from a long time ago and how has this relationship evolved over time" was that bats were not a common feature around homesteads in the past. Complementary explanations on historically fewer human-bat interactions were offered, including, 'human population was low, settlements were scattered, and undisturbed landscapes were common', providing space for bats and other wild animals to live largely undisturbed from human encroachment. Accordingly, an increased presence of bats has been noticed in human communities over

the last two decades. This notion is consistent with nearly twofold increase in human population size in the country, 2000 to 2020 [41,53]. To what extent increases in fruit production, which is largely done by smallholders, influenced the abundance and proximity of fruit bat colonies needs to be analyzed in the future. Local research into bat-mediated ecosystem services including pollination, seed dispersal and pest suppression [54,55] and potential damage to the harvest [56,57] is needed for this region and could provide the foundation for evidence-based and protracted sensitization campaigns.

The prevailing opinion of most of the participants was that bats should not live within homes or immediate human environments. This proclamation is substantiated by numerous actions of communities aimed at disrupting the bat roosting resources and roost sites. The recommendation of the elderly participants in the study area that bats, and humans should not share the same dwellings is apparently linked to previous bad experiences of sharing dwellings. For instance, homesteads neighboring *E. helvum* colonies in Moyo district did not want the bats in their neighborhood because they felt inconveniences from the noise, smell, and specifically from the perceived spread of bedbugs and bat lice, in addition to fear of diseases from bats and destruction of fruits. The reported increase in the presence of bats in human communities over the recent past and occasional violent responses, such as destroying the bat roosting resources, is a cause of concern. Whether this increased bat-human proximity is due to loss of natural habitats or to the synanthropic nature of some bat species as they respond to resource availability (food and/or roosting sites), or both, is unknown—but the consequences are the same [58–61]. Some species may be regionally extirpated with habitat loss, lowering overall diversity, while others adapt well to encroachment, raising spillover risk to humans.

### 4.4. Willingness of Local People to Protect and Conserve Bats

Almost equal proportions of participants in our study expressed willingness and unwillingness to protect and conserve bats. A study in Namibia indicated that negative cultural representations of bats and positive attitudes towards bats (over 95% agreed that bats should be conserved) are disconnected [46], which is also implied in this study, given the relatively high number of respondents that are willing to conserve bats despite the negative myths and attributes assigned to bats. We note that, despite limited knowledge to back their support for bat conservation, the pro-bat conservation respondents argued that every creature renders some service in nature, whether such services are known or unknown by humans, and that every creature has an inherent right to live, so, their fate to exist should not be interpreted in anthropocentric terms. This mindset can be taken advantage of as it lays the foundation upon which a locally adapted sensitization campaign could build, emphasizing the ecosystem services provided by regional bats and the common interest to protect these species (e.g., [62]). On the other hand, those opposed to bat conservation presented the negative aspects about bats including the role of bats in the spread of dangerous diseases as was also reported e.g., from studies in Kenya and Madagascar [24,26]. In contrast, respondents in studies in Namibia and Ghana rarely associated bats with diseases [46,47]. Animosity to bats was based on ugliness, smell, spread of bedbugs and bat lice, destruction of fruits, and fears arising from negative perceptions from myths and folktales which has been reported from other studies in Africa and Asia (e.g., [24,63,64]). Understanding of the specific local conflicts between bats and humans, e.g., destruction of agricultural fruits, is needed to design appropriate interventions [63]. Despite these negative perceptions, we also observed that people in rural areas had a low tendency to destroy bat roosting resources as compared to those in urban areas. This may be because urban communities more readily access the negative scientific and media reporting and attribution of emerging diseases to bats, thus worsening the already bad reputation of bats and scaring urban communities into destroying bat roost sites in their localities. The low tendency to destroy bat roosting resources in rural areas despite prevailing negative myths is consistent with a study [25] that showed that threatened and endangered bat species have thrived in territories where local people have exclusive land rights. This

may also be due to inherent biophilia that is disrupted when people are removed from nature [65,66]. The study area is characterized by high rates of deforestation, and the tendency of fruit bats to shift their colonies to peri-domestic environs, or vice versa to not leave an area when people encroach into their original habitats, is increasing [67,68]. Roost site destruction only happens when roosts are in houses or occasionally when bats roost in the trees of home compounds (especially for gregarious and often noisy species). Like the great African environmentalist Baba Dioum, we argue that people protect and conserve what they understand and value. Because bats are not well understood and less or not valued across all communities we interacted with, the indifference expressed towards them in some communities was expected. This view is supported by the findings [69] that public unawareness of the numerous ecosystem services provided by wildlife limits their willingness and support to conservation efforts. There is an urgent need to create awareness and develop evidence-based measures to protect bats and their habitats from the increasing threats of degradation and habitat loss [70,71].

### 4.5. Conservation Action and Impact Surveys

From our rapid sensitization campaign that prevented persecution of bats, we realized that sustained conservation requires convincing evidence. However, actions should not wait for scientific evidence where immediate danger is obvious. Many critical species and habitats of conservation concern are found in private or communal lands without any form of protection or baseline data. Bats in Uganda and elsewhere are regularly regarded as a single species, with many people uninformed about the fact that bats are an exceptionally species-rich group of mammals. Bats get condemned for the potential threats posed by a few species. Many studies have found that the public perception of bats is predominantly negative because of lack of understanding of their ecological and economic roles [49,63,72] and that this gap of knowledge deficiency is due to lack of prioritization and funding to support bat conservation [73]. More studies need to focus on understanding the quality of bat habitats and the underlying survival mechanisms in fast-changing human environments.

One key strategy to effectively counter public misconceptions is by raising positive perceptions of the focal species or habitat in that region [64,74,75]. There are various outlets to improve public perception toward species and promote wildlife conservation, including zoos, aquariums, and wildlife watching [75], but these resources are mainly valued by and affordable to foreign visitors. These initiatives are widely known to promote "charismatic species", e.g., large mammals, felids, and birds [76,77], but are very scarce for unfamiliar species like bats. Conservation education and outreach with communities have been shown to improve public understanding and acceptance of bats [74]. Flying fox viewing in Terengganu, Malaysia showed evidence of the potential of wildlife tourism as a conservation tool by raising public awareness and generating positive attitudes towards bats [78]. Like other wildlife tourism initiatives, bat-watching has the potential to raise awareness about bats and to promote sustainable local development, therefore facilitating their conservation if conducted effectively and with appropriate sensitivity. However, there is limited knowledge on the effectiveness of bat-watching and its ability to improve human-bat relations in many parts of the world, particularly in the global south [63,79].

Based on our survey results, we recommend deliberate and sustained community sensitizations about bats through school programs for young people, and promotion of safety-distanced bat-watching tourism and bat education at roosting sites. Important additional steps in Northern Uganda could include educational signage near known sites in need of protection, information about safe practices for excluding bats from buildings, and protection of existing roost sites to increase the quality of roosts and minimize encounters with humans. Because older people had more positive attitudes toward bats than youth, bat education awareness that targets young people and includes participation of the elderly is desirable. Published work suggests that when adult environmentalists were asked about the origin of their commitment to protect the environment, most mentioned positive

experience with nature during childhood [80–82]. Accordingly, environmental educators have consequently stressed activities that increase children's contact with nature [83–85]. Findings from a study conducted in Arabuko-Sokoke Forest (ASF) in Kenya show that evening school-based bat study tours may be an effective approach because they would involve catching bats using mist-nets, showing features of bats to learners, answering questions from students, and explaining details about bats using hand-held live bats [29]. Safe educational encounters with live bats, in which bat scientists wear appropriate PPE (e.g., masks and gloves [86] for both human and bat protection) may positively influence human attitudes toward bats [87,88].

## 5. Conclusions

The study revealed some long-established, negative perceptions of bats in communities in Northern Uganda, recently exacerbated by the SARS-CoV-2 pandemic. No single deliberate conservation efforts exist in the entire study area and the negative perception largely hampers the implementation of bat conservation in Northern Uganda. The prevailing picture is that people in the study area are not yet prepared to accept coexistence with bats, due to deep-seated beliefs or myths they have been holding. Continuing discussions and sensitizations are recommended to realize a sustainable change in attitudes to and coexistence with bats. We argue that documenting and understanding what different communities know about bats to understand the myths or stories of each community that shape their attitudes towards bats will aid bat conservation efforts on a regional and local scale. Knowledge of social norms and what motivates peoples' actions toward bats can inform the design of messages aimed at influencing appropriate behavior towards bats [23]. The SARS-CoV-2 pandemic has triggered atrocities against bats and presents considerable challenges for bat conservation. More studies on human attitudes towards bats under the current circumstances are warranted as the framing of messages that link bats to the direct transmission of SARS-CoV-2 to humans can foster false beliefs of their role in disease outbreaks and consequently influence the willingness of people to coexist with bats [89–95]. Reframing this narrative is critical.

The emergency community sensitization and conservation steps taken in the middle of the SARS-CoV-2 pandemic in West Nile Districts in Uganda provided the local communities with the evidence needed to combat the prevailing negative narratives about bats, changed some decisions, and very specifically saved two bat roost sites from destruction. This demonstrates that bat fact-based sensitization and community action can be effective tools to protect bats. Deliberate and continuous community sensitizations studies in Northern Uganda are essential to implement first conservation steps. We further recommend creative introduction of bat education to the young in schools as well as targeted information transfer to communities. These could be facilitated by the creation of a dedicated organization for bat conservation in Uganda with the aim to gradually develop innovative programs for protracted, wide-spread, and affordable conservation programs that benefit bats, their habitats, and humans alike.

**Supplementary Materials:** The following supporting information can be downloaded at: https://www.mdpi.com/article/10.3390/su142416924/s1, Text S1: Questionnaire—Knowledge and awareness about bats in Northern Uganda; Table S1: Overview responses of individual and group interviews; Table S1(A) Respondents count (n) and frequency (%) of the responses for questions regarding basic knowledge about bats (Answers YES/NO); Table S1(B) Respondents count (n) and frequency (%) of the responses for questions regarding basic knowledge about bats (open questions); Table S2: Interviews elderly; Text S2: Extended results—sensitization campaign.

**Author Contributions:** Conceptualization, I.E., J.S., R.K., K.M. and D.M.R.; methodology, I.E., J.S. and D.M.R.; software, J.S.; validation, I.E., D.M.R. and J.S.; formal analysis, I.E. and J.S.; investigation, I.E.; resources, J.S. and D.M.R.; data curation, I.E. and J.S.; writing—original draft preparation, I.E.; writing—review and editing, I.E., J.S., D.M.R., K.M. visualization, I.E. and J.S.; supervision, J.S.; project administration, J.S., D.M.R.; funding acquisition, J.S. and D.M.R.; All authors have read and agreed to the published version of the manuscript.

**Funding:** I.E. is supported by a Ph.D. scholarship of the German Academic Exchange Service (DAAD), J.S. is funded by an individual research grant from the German Research Foundation (DFG; project number 437846632), D.M.R. is supported by the US National Institute of Allergy and Infectious Diseases (NIAID) (1R01AI151144).

**Institutional Review Board Statement:** The study was conducted in accordance with the Declaration of Helsinki and approved by the Ethics Committee of Muni University (May 2020) and Uganda National Council for Science and Technology UNCST (Permit No. NS667 issued in February 2020). Permission was also sought and granted for all photos of study participants used here.

**Informed Consent Statement:** Informed consent was obtained from all subjects involved in the study.

**Data Availability Statement:** All data supporting reported results are included in the manuscript and the Supplementary Materials.

**Acknowledgments:** We thank the 151 participants who accepted to share their views in the study. We appreciate the administrative, technical, and political offices and officials in the districts of Moyo, Koboko and Yumbe districts, and the management of health and education institutions in the study area that permitted their community to participate in the study. We appreciate the field team of Adiga Kassim, Saidi Omar Dramani and Anguyo Dennis Foe for their immense help during data collection. We thank EcoHealth Alliance and partners for permitting the use and adaptation of the booklet "Living Safely with Bats" that was developed by EcoHealth Alliance and partners as a resource to help prevent spread of zoonotic disease carried by bats (PREDICT One Health Consortium 2018).

**Conflicts of Interest:** The authors declare no conflict of interest.

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
