# Peer review of "Negative Perception of Bats, Exacerbated by the SARS-CoV-2 Pandemic, May Hinder Bat Conservation in Northern Uganda"

_sustainability, doi:10.3390/su142416924_

Round 1

Reviewer 1 Report

The paper presents interesting results that are valuable to the human-bat interface in high-bat biodiversity regions. The study provides new insights into how pre- and during the COVID-19 pandemic affect human perception and willingness for bat conservation and protection. However, the current form of the paper contains an enormous information and a lengthy write-up that I think diluted the overall value of the paper. The manuscript contains numerous redundant statement and mismatched ideas that is exhausting to read and follow. I suggest to the authors to reduce the content of the paper, merge similar and related ideas, and focus on what is the key objective of the paper.

Secondly, the manuscript has no clear and defined objectives which do not support the overall conclusion of the paper. The paper also needs a strong and succinct conclusion.

Overall, the paper needs more work and organisation of the narratives from the locals. Please see the attached manuscript for my specific comments.

Author Response

REVIEWER 1

The paper presents interesting results that are valuable to the human-bat interface in high-bat biodiversity regions. The study provides new insights into how pre- and during the COVID-19 pandemic affect human perception and willingness for bat conservation and protection.

We thank the reviewer for the supportive statement on our research topic.

However, the current form of the paper contains an enormous information and a lengthy write-up that I think diluted the overall value of the paper. The manuscript contains numerous redundant statement and mismatched ideas that is exhausting to read and follow. I suggest to the authors to reduce the content of the paper, merge similar and related ideas, and focus on what is the key objective of the paper.

Thank you for your constructive comments and for your many helpful markups in the document. We have followed your suggestions as noted below. The manuscript is now much shorter, and focused. Especially the introduction has been extensively shortened and now only focuses on the topics of our objectives.

Secondly, the manuscript has no clear and defined objectives which do not support the overall conclusion of the paper. The paper also needs a strong and succinct conclusion. Overall, the paper needs more work and organisation of the narratives from the locals. Please see the attached manuscript for my specific comments.

As above, we have followed your suggestion and improved clarity, including by better defining clear objectives. We now define the objectives at the end of the introduction: “Our objective in this study was to document existing knowledge about and perceptions of bats in Northern Uganda among different local Ugandan communities and to describe differences in viewpoint and knowledge between elders and younger study participants. Additionally, we present the results of a sensitization campaign about bats in Northern Uganda that prevented planned actions against bats that resulted from COVID-19 messaging.”

The conclusion statements have been shortened and specify the implications of the study and research findings.

Reviewer 2 Report

I found the manuscript interesting for the topic dealt with. The article also provides useful and documented information on the interactions between humans and wild species and how these can be affected by prejudices, contingent conditions and deterioration of ecosystems.

The introduction contextualizes well the subject and gives appropriate starting scientific basis, but the objective of the study seems too broad and perhaps should be better defined and specified. In the abstract, for example, I read (lines 14-16) that the study was conducted in response to encounters with local communities ..., but the research question to be answered is not specified. The objective should also be better defined in the introduction where I read (lines 114-116) that the results on the topic are presented, but the research gap and the purpose of the study are not clearly and directly explained.

Materials and methods

(lines 123-128) the phrase should be included in the introduction and only briefly resumed in the methodology. Furthermore, it would be useful to include a list of the subsequent paragraphs to help the reader to understand the structure of the methodology.

In Survey logistics, I did not understand whether the choice of doing two types of interviews, individual and group, derives from SARS COV-2 containment measures or from a methodological need or ,even, to bring out particular aspects from respondents. If so, it would be better to highlight it in the results as well.

Explain the follow-up sensitization campaign carried out in February 2021. This is cited only in the results but not in the methodology. Did I miss it?

Results

I think the results can be simplified slightly to allow for a fluid and more effective reading. Since most of the findings are of a qualitative nature, a table that schematizes at least the descriptive statistics of the quantitative aspects could help to understand them better.

Discussion

The discussion effectively contextualizes the results obtained and strive to compare the results with other studies, although the purely descriptive nature of the survey does not help to compare the findings with other similar research. It would be appropriate to take as references other studies i.e. on community preferences and the willingness to accept coexistence with wild species or to protect biodiversity even at the expense of agricultural production.

Conclusions.

The conclusions could be better directed to specify the implications of the study and research findings.

Author Response

REVIEWER 2

I found the manuscript interesting for the topic dealt with. The article also provides useful and documented information on the interactions between humans and wild species and how these can be affected by prejudices, contingent conditions and deterioration of ecosystems.

Thank you for your constructive comments. Please see our reply to your comments after each of the following sections.

The introduction contextualizes well the subject and gives appropriate starting scientific basis, but the objective of the study seems too broad and perhaps should be better defined and specified. In the abstract, for example, I read (lines 14-16) that the study was conducted in response to encounters with local communities ..., but the research question to be answered is not specified. The objective should also be better defined in the introduction where I read (lines 114-116) that the results on the topic are presented, but the research gap and the purpose of the study are not clearly and directly explained.

We have clarified the objectives and removed text that was confusing. The introduction is now much shorter (3 paragraphs) and focused. Thank you for your very helpful suggestions.

Materials and methods

(lines 123-128) the phrase should be included in the introduction and only briefly resumed in the methodology. Furthermore, it would be useful to include a list of the subsequent paragraphs to help the reader to understand the structure of the methodology.

The sentiments in statements have been shifted to the discussion. This section is now more concise, and we feel that the existing subheading structure is sufficient to enable the reader to understand.

In Survey logistics, I did not understand whether the choice of doing two types of interviews, individual and group, derives from SARS COV-2 containment measures or from a methodological need or ,even, to bring out particular aspects from respondents. If so, it would be better to highlight it in the results as well.

Focus groups and individual questionnaires can serve different purposes and facilitate information gathering. As now clarified in the text, “Individual interviews were complemented with focus group interviews for the following reasons: (1) Participants were more likely to consent to participate in a group setting; (2) the group discussions setting facilitated contributions and thus information; (3) movement of both the interviewers and potential participants was limited during the SARS-COV-2 pandemic lockdown and group interviews allowed involvement of more participants in each area.”

Explain the follow-up sensitization campaign carried out in February 2021. This is cited only in the results but not in the methodology. Did I miss it?

We actually carried out a follow-up sensitization campaign with interviews in February 2021. However, unfortunately the material was lost in the field and could therefore not be analysed. However, we still gathered information about the positive impacts of the sensitization campaign in 2020 (which prevented the planned actions against bats) by assessing the situation in the affected areas in February 2021. The sentence that we “conducted a follow-up survey” was not appropriate and should not have been in the first version of the manuscript and we now deleted it accordingly.

Results

I think the results can be simplified slightly to allow for a fluid and more effective reading. Since most of the findings are of a qualitative nature, a table that schematizes at least the descriptive statistics of the quantitative aspects could help to understand them better.

We aimed at simplifying the results section and improving the flow of the consecutive sections. We also revised and summarised our qualitative findings in tables that are now part of the main manuscript (have been hidden in the supplement before).

Discussion

The discussion effectively contextualizes the results obtained and strive to compare the results with other studies, although the purely descriptive nature of the survey does not help to compare the findings with other similar research. It would be appropriate to take as references other studies i.e. on community preferences and the willingness to accept coexistence with wild species or to protect biodiversity even at the expense of agricultural production.

This is an important point, and we aimed to compare our study findings to published literature and added and discussed additional studies.  

Conclusions.

The conclusions could be better directed to specify the implications of the study and research findings.

We have followed your suggestion and improved the conclusion statements.

Reviewer 3 Report

The paper describes a very interesting and relevant issue. The article is original, has good technical quality, and high general interest. However, the Data analysis is rather poor and should be expanded

Author Response

REVIEWER 3

The paper describes a very interesting and relevant issue. The article is original, has good technical quality, and high general interest. However, the Data analysis is rather poor and should be expanded

We thank the reviewer for highlighting the importance and relevance of our study. We were also happy to read that our study is original, technically sound and of broad interest to the public. We understand that our data analysis was below the reviewer’s expectations and should be improved. Since the statement is overly brief and vague, we can only guess what the reviewer refers to. In our internal discussions during the review process, we spent considerable time to improve our data analysis, and organised results into tables that we think will be helpful. Accordingly, we hope that we also addressed this reviewer’s concerns.

Reviewer 4 Report

Review of sustainability-1948023

This manuscript documents the knowledge and perception/attitudes toward bats from interviews (n = 151 participants) in northwestern Uganda. The paper addresses an important knowledge gap of how bats were perceived in East Africa during the COVID-19 pandemic. Overall, this work was great and definitely deserves to be published, but I have some suggestions that I hope the authors can address prior to publication.

1.     My main concern is that the interview responses were generalized to groups of people throughout the Results (e.g., Section 3.4). Given the limited number of interviews and the use of semi-structured interviews (often used to gain a more in-depth knowledge of how a subset of people respond on topics), it would be better to limit your interpretation to this is how the respondents felt rather than how an entire ethnic group felt. For instance, cultural stories weren’t given across all respondents of a certain ethnic group (more common instead among older respondents).

2.     One other main point that I feel the authors could address is providing more ecological information on the bat community of this area. How many species are present? Are people more likely to interact/be familiar with fruit bats vs. insectivorous bats? Are the bats that roost in homes only insectivorous species? Etc.

Specific comments:

1.     Introduction, Paragraph 1: I believe the S in “white-nose Syndrome” should be lower-case

2.     Materials and Methods: Were the interviews recorded or were notes taken throughout the interview?

3.     Materials and Methods, Study area Paragraph 2: I suggest rewording “and from random people living in urban and rural areas” to “and from random encounters with people living in urban and rural areas”

4.     Materials and Methods, Study area Paragraph 2: No need to capitalize the cohort groups at the end of this paragraph

5.     Materials and Methods, Individual Interviews Paragraph 1: “thirty-five persons” may be better written as “thirty-five individuals”

6.     Figure 3: “Campaigns” is spelled incorrectly in panel C

7.     3.2, Paragraph 3: “Mirred” should be “mirrored” I think?

8.     3.5, Paragraph 1: “The shades of the mango trees” should be “The shade of…”

9.     Discussion, Paragraph 1: Statement too strong “This finding raises the possibility that in homes there are no discussions about nature and that transmission of indigenous knowledge about the environment to the young people no longer occurs.” Just because young respondents didn’t know much about bats doesn’t mean that there are no discussions about nature. Instead they may simply not discuss bats.

10.  Discussion, Willingness of Local People Paragraph 1: Odd placement of e.g, in middle of sentence “This thinking can provide an excellent foundation for locally adapted sensitization campaigns that should e.g. highlight ecosystem services provided by bats specifically in the study area and the common interest to protect these bats”. I suggest removing or replacing with “for example”

11.  Discussion, Willingness of Local People Paragraph 1: “The low tendency to destroy bat roosting resources in rural areas despite prevailing negative myths is consistent with the findings of Fernandez-Llamazares et al. ([25])” is also found in other studies through Africa. You may want to make some additional comparisons to other studies in the Discussion. For instance, Laverty et al. (2021) found largely positive attitudes despite negative myths among Namibian pastoralists (https://doi.org/10.2993/0278-0771-41.1.70). In contrast to this study, over 90% of their respondents supported conservation efforts for bats.

12.  Some typos are present late in the Discussion. I suggest doing a quick re-read there.

Author Response

This manuscript documents the knowledge and perception/attitudes toward bats from interviews (n = 151 participants) in northwestern Uganda. The paper addresses an important knowledge gap of how bats were perceived in East Africa during the COVID-19 pandemic. Overall, this work was great and definitely deserves to be published, but I have some suggestions that I hope the authors can address prior to publication.

We are grateful to the reviewer for this encouraging and frank assessment. We addressed all of the suggestions made by Reviewer 4 which has improved the manuscript.

  1. My main concern is that the interview responses were generalized to groups of people throughout the Results (e.g., Section 3.4). Given the limited number of interviews and the use of semi-structured interviews (often used to gain a more in-depth knowledge of how a subset of people respond on topics), it would be better to limit your interpretation to this is how the respondents felt rather than how an entire ethnic group felt. For instance, cultural stories weren’t given across all respondents of a certain ethnic group (more common instead among older respondents).

The reviewer raises an important point. In the revised version we removed statements that generalized across different ethnic groups and limited our interpretation to how the respondents felt, as suggested. We recognize that follow-up work is warranted to further validate and expand our present findings.

  1. One other main point that I feel the authors could address is providing more ecological information on the bat community of this area. How many species are present? Are people more likely to interact/be familiar with fruit bats vs. insectivorous bats? Are the bats that roost in homes only insectivorous species? Etc.

We thank Reviewer 4 for pointing out this important issue. However, no publications/data are available for the information about the ecology, occurrence, diversity of bat species in the area. We added the following sentence in the method part: "Despite their huge representation in Uganda’s mammalian fauna, bat diversity surveys have been conducted in very few areas and the distribution of bats in the country is poorly known [31,40] and no data has been published from the West Nile region to date."

From our interviews, people only knew bats as big versus small, house versus tree bats etc. without any further details. We added the following sentence in the result part: "According to the descriptions by the respondents, people are bothered by fruit bats in trees (most probably epauletted fruit bat species, e.g. Epomophorus spp. and Eidolon helvum) and insectivorous bats that roost in their houses (most probably molossid bats)."

We haven't done any ecological surveys ourselves yet as our focus so far has been on fruit bats and bat health. In general, information on bat diversity and ecology in Uganda is very scarce which highlights the need to perform longitudinal studies towards a tempo-spatial resolution of bat species distribution and bat-human interactions. Given the editorial request to substantially shorten the manuscript we felt that addressing these questions in the introduction in a largely speculative manner will be inadequate.

Specific comments:

  1. Introduction, Paragraph 1: I believe the S in “whitenose Syndrome” should be lower-case

Corrected as suggested.

  1. Materials and Methods: Were the interviews recorded or were notes taken throughout the interview?

The interviewer recorded all information on the printed questionnaires which we now also state in the text.

  1. Materials and Methods, Study area Paragraph 2: I suggest rewording “and from random people living in urban and rural areas” to “and from random encounters with people living in urban and rural areas”

We thank the reviewer for pointing out this poor misleading phrasing and have changed the wording as suggested.

  1. Materials and Methods, Study area Paragraph 2: No need to capitalize the cohort groups at the end of this paragraph

We agree.

  1. Materials and Methods, Individual Interviews Paragraph 1: “thirty-five persons” may be better written as “thirty-five individuals”

Agreed.

  1. Figure 3: “Campaigns” is spelled incorrectly in panel C

This has been corrected now.

  1. 3.2, Paragraph 3: “Mirred” should be “mirrored” I think?

Thanks, this has been changed now

  1. 3.5, Paragraph 1: “The shades of the mango trees” should be “The shade of…”

Yes, indeed.

  1. Discussion, Paragraph 1: Statement too strong “This finding raises the possibility that in homes there are no discussions about nature and that transmission of indigenous knowledge about the environment to the young people no longer occurs.” Just because young respondents didn’t know much about bats doesn’t mean that there are no discussions about nature. Instead they may simply not discuss bats.

We agree that this statement is too strong as we cannot support it with data from our study. However, the lead author´s experience (who lives in this area) and his conversations with the interviewees indicate that there are no discussions about nature to young people in their homes. Every learning is delegated to the formal school systems that hardly tackle indigenous knowledge. This is currently a big societal concern in the study area. However, as there is no published data to support this, we deleted the statement.

  1. Discussion, Willingness of Local People Paragraph 1: Odd placement of e.g, in middle of sentence “This thinking can provide an excellent foundation for locally adapted sensitization campaigns that should e.g. highlight ecosystem services provided by bats specifically in the study area and the common interest to protect these bats”. I suggest removing or replacing with “for example”

Changed as suggested

  1. Discussion, Willingness of Local People Paragraph 1: “The low tendency to destroy bat roosting resources in rural areas despite prevailing negative myths is consistent with the findings of Fernandez-Llamazares et al. ([25])” is also found in other studies through Africa. You may want to make some additional comparisons to other studies in the Discussion. For instance, Laverty et al. (2021) found largely positive attitudes despite negative myths among Namibian pastoralists (https://doi.org/10.299/0278-0771-41.1.70). In contrast to this study, over 90% of their respondents supported conservation efforts for bats.

We thank Reviewer 4 for highlighting other studies that found positive attitudes despite negative myths which is really interesting and important. We therefore agree that an additional comparison to other studies is warranted. However, in order to shorten the manuscript by 25%, we decided to not enlarge the discussion (and to delete our sentence) on this particular point which requires a thorough discussion.

  1. Some typos are present late in the Discussion. I suggest doing a quick re-read there.

Thanks for pointing this out. This has been checked now.

Round 2

Reviewer 1 Report

I appreciate the authors' effort to revise their manuscript. I have no further comments.

Author Response

We thank Reviewer 1 for the helpful suggestions and comments in the first revision round.

Reviewer 2 Report

The manuscript has been significantly improved thanks to a clearer structure and a more concise and direct definition of the objectives, methodology and presentation of results.

Author Response

We thank Reviewer 2 for the positive assessment of our manuscript and are thankful for the helpful suggestions and comments in the first revision round.

Reviewer 4 Report

I thank the authors for addressing my earlier concerns and I feel that the manuscript is much improved.

My only outstanding concern is that the Discussion appears to largely be about the paper's own results. If the authors could frame it more around how the finding of their study compare to that of other studies in Africa or in similar habitats (there's a whole special issue on the ethnobiology of bats in the Journal of Ethnobiology for instance), then I think readers could gain a lot more from this study. I recognize that the editor requested that the manuscript length be reduced in the last round of revision, so I'll let them make the call on this. 

Additionally, there continues to be minor typos throughout the paper, but those will hopefully be addressed in the copyediting stage.

Once again, I congratulate the authors on their contribution to the field and I look forward to seeing this paper come out. 

Author Response

I thank the authors for addressing my earlier concerns and I feel that the manuscript is much improved.

We thank reviewer 4 for the positive assessment of our revised manuscript.

My only outstanding concern is that the Discussion appears to largely be about the paper's own results. If the authors could frame it more around how the finding of their study compare to that of other studies in Africa or in similar habitats (there's a whole special issue on the ethnobiology of bats in the Journal of Ethnobiology for instance), then I think readers could gain a lot more from this study. I recognize that the editor requested that the manuscript length be reduced in the last round of revision, so I'll let them make the call on this. 

Thank you for raising this concern. We agree that we should have discussed our results more in context with similar studies and apologize for this oversight. In the attempt to shorten the overall manuscript, the comparison of our results to published studies fell short.

We have now revised our manuscript by discussing some of our results in context with similar studies that have been carried out in Africa and Asia. We feel that we cannot compare all results to the findings in other studies due to the sometimes very different study designs. However, where possible, we compared and discussed our results to similar studies. We kept the edits brief to still meet the requirements of the last revision round of shortening the overall manuscript.

Additionally, there continues to be minor typos throughout the paper, but those will hopefully be addressed in the copyediting stage.

We went through the whole manuscript and corrected the typographical errors.

Once again, I congratulate the authors on their contribution to the field and I look forward to seeing this paper come out. 

Thank you very much. Our manuscript has definitely benefited from your reviews, and we thank you for your time and effort.